# Structural rearrangements allow nucleic acid discrimination by type I-D Cascade

Evan A. Schwartz[1,2,8], Tess M. McBride [3,8], Jack P. K. Bravo[1,8], Daniel Wrapp [1,2], Peter C. Fineran [3,4,5], Robert D. Fagerlund [3,4,5] & David W. Taylor [1,2,6,7✉]

CRISPR-Cas systems are adaptive immune systems that protect prokaryotes from foreign nucleic acids, such as bacteriophages. Two of the most prevalent CRISPR-Cas systems include type I and type III. Interestingly, the type I-D interference proteins contain characteristic features of both type I and type III systems. Here, we present the structures of type I-D Cascade bound to both a double-stranded (ds)DNA and a single-stranded (ss)RNA target at 2.9 and 3.1 Å, respectively. We show that type I-D Cascade is capable of specifically binding ssRNA and reveal how PAM recognition of dsDNA targets initiates long-range structural rearrangements that likely primes Cas10d for Cas3′ binding and subsequent non-target strand DNA cleavage. These structures allow us to model how binding of the anti-CRISPR protein AcrID1 likely blocks target dsDNA binding via competitive inhibition of the DNA substrate engagement with the Cas10d active site. This work elucidates the unique mechanisms used by type I-D Cascade for discrimination of single-stranded and double stranded targets. Thus, our data supports a model for the hybrid nature of this complex with features of type III and type I systems.

[1] Department of Molecular Biosciences, University of Texas at Austin, Austin, TX 78712-1597, USA. [2] Interdiscplinary Life Sciences Graduate Programs, University of Texas at Austin, Austin, TX 78712-1597, USA. [3] Microbiology and Immunology, University of Otago, PO Box 56 Dunedin 9054, New Zealand. [4] Bioprotection Aotearoa, University of Otago, PO Box 56 Dunedin 9054, New Zealand. [5] Genetics Otago, University of Otago, Dunedin, New Zealand. [6] Center for Systems and Synthetic Biology, University of Texas at Austin, Austin, TX 78712-1597, USA. [7] LIVESTRONG Cancer Institutes, Dell Medical School, Austin, TX 78712-1597, USA. [8] These authors contributed equally: Evan A. Schwartz, Tess M. McBride, Jack P. K. Bravo. ✉email: dtaylor@utexas.edu

The evolutionary arms race between prokaryotes and their viral invaders has led bacteria and archaea to evolve defense mechanisms[1]. CRISPR (clustered regularly inter-spaced short palindromic repeats)-Cas (CRISPR-associated) systems provide adaptive immunity by targeting foreign nucleic acids and mobile genetic elements in a sequence-specific manner[2]. During CRISPR-Cas immunity, fragments of invading nucleic acids are stored as spacers in the host CRISPR loci[3,4]. Spacers are transcribed and processed into mature CRISPR RNA (crRNA) guides, which assemble with Cas proteins and form CRISPR-Cas surveillance complexes. The CRISPR-Cas surveillance complexes use the crRNA as a guide to bind complementary nucleic acid sequences and trigger their degradation[5]. CRISPR-Cas complexes are grouped into 2 main classes, 6 types, and >35 sub-types[6,7]. The two classes are characterized by either containing a multi-subunit effector complex (class 1) or a single-protein ribonucleoprotein complex (class 2). Class 2 complexes are the most widely studied due to their current application as genome editing tools[8,9]. However, class 1 systems account for ~80% of all CRISPR-Cas systems in bacterial and archaeal genomes[7] and are starting to be established as tools for bio-technological applications[10–12].

Class 1 CRISPR systems comprise type I, III, and IV systems[6]. Type I effector complexes, Cascades (CRISPR-associated complex for antiviral defense), target protospacer-containing DNA that is complementary to the crRNA. These systems are classified by the presence of the signature protein Cas3, a nuclease-helicase sub-unit that acts in trans following recruitment by Cascade[13,14]. Type I systems use a short protospacer adjacent motif (PAM) to discriminate between self and non-self DNA. PAM recognition by the large subunit Cas8 is then followed by base-pairing between the protospacer and crRNA, displacing the non-target strand and creating an R-loop. R-loop formation facilitates conformational rearrangements and recruitment of Cas3 to Cas8 for processive degradation of dsDNA targets[15–17]. In contrast, type III systems bind to RNA complementary to their crRNA where in many cases Cas7 subunits cleave the RNA. It is hypothesized the type III signature protein, Cas10, may induce non-specific ssDNA nicks at the transcription bubble[18–21]. Upon activation of Cas10, a Palm domain synthesizes cyclic oligoadenylate (cOA) that in turn activates downstream nuclease enzymes[22,23]. Interestingly, the type I-D system utilizes subunits that are homologs of type I complexes, type III complexes, or both[24–27]. The type I-D operon contains the type I signature gene *cas3* split into its two domains; the helicase domain (Cas3′) encoded from a stand-alone gene, and the HD nuclease domain (Cas3″), which is fused to the large subunit as part of the core effector complex. The Cas10d subunit is bigger than other type I large subunits (Cas8) and similar to type III large subunits[24,25,28], yet retains the type I functionality of discerning PAMs in target DNA sequences[29,30].

Our previous studies have shown that type I-D Cascade utilizes non-canonical small subunits (Cas11d) that are alternatively translated from the 3′ end of *cas10d* and likely facilitate R-loop formation during target recognition[25]. Internal translation of small subunits is widespread, occurring in ~23% of all known CRISPR-Cas systems. Recently, our structural and biochemical studies on the type I-C Cascade highlighted how non-canonical small subunits are incorporated into type I complexes[25,31].

While several Cascade structures have been determined to date[31–33], a high-resolution structure of the type I-D Cascade was lacking. The type I-D system has been previously shown to utilize a 5′-GTN-3′ PAM upstream of the protospacer on the non-target strand for CRISPR adaptation and to elicit interference[29,30,34,35]. However, the underlying mechanisms of how this type I/III hybrid effector complex targets dsDNA are not fully understood. To address this, we determined the structure of type I-D Cascade bound to dsDNA and show critical interactions for PAM recognition. We further showed this hybrid Cascade can specifically bind ssRNA and determined the structure of this complex. Here, we show that dsDNA induces significant structural rearrangements to accommodate the non-target DNA strand and position the nucleic acid appropriately for cleavage by the Cas10d nuclease domain.

## Results

**Cryo-EM structure of type I-D Cascade.** Previously, we demonstrated that type I-D Cascade specifically bound to a complementary dsDNA protospacer substrate containing a 5′-GTT-3′ PAM with an apparent dissociation binding constant of $35 \pm 3$ nM[25]. To obtain mechanistic insight into dsDNA targeting, we employed cryo-electron microscopy to directly visualize type I-D Cascade bound to dsDNA with a protospacer sequence complementary to the crRNA and flanked by a GTT PAM (Supplementary Fig. 1). Raw micrographs and reference-free 2D class averages showed particles with a "sea-worm"-like shape that are 230 Å along the longest dimension (Supplementary Figs. 2 and 3). We employed focused 3D classification to overcome local conformational heterogeneity (Supplementary Fig. 2), which yielded a structure of the dsDNA-bound type I-D Cascade at a global resolution of 2.9 Å and enabled de novo atomic modeling of the entire complex (Fig. 1 and Supplementary Fig. 4). The stoichiometry of the complex was $Cas10d_1{:}Cas7d_7{:}Cas5d_1{:}Cas11d_2$, highlighting the addition of one Cas7d subunit compared to our recent sub-nanometer structure. However, the density of the Cas7d subunit is slightly more ambiguous, suggesting heterogeneity in the length of the Cas7d filament. Due to flexibility and/or compositional heterogeneity at the top of the complex, we were unable to resolve Cas6d in this high-resolution structure, despite being the same sample where Cas6d was observed in the purified complex via SDS-PAGE and mass spectrometry[25].

As is typical for Cascade complexes, a repeating, helical backbone of Cas7d subunits assembles around the crRNA, while the C-terminus of Cas10d and two Cas11d subunits constitute a minor filament along the belly of the complex (Fig. 1). At the base of the complex, Cas5d recognizes the 5′ eight nucleotide handle of the crRNA. Adjacent to Cas5d is Cas10d, which resembles a cowboy boot with protruding spur-like and toe-like domains. This spur interacts with the duplex of the double-stranded DNA target. The conformation of the crRNA and Cas7d backbone appears to be more like that of type III than that of type I effector complexes, with a near-vertical path of the crRNA (Supplementary Fig. 5). Due to the near-identical crRNA and Cas7d filament geometry, the overall shape of type I-D Cascade aligns better with type III than type I complexes, highlighting its potential as an evolutionary intermediate between type III and type I CRISPR-Cas[25]. Of all known type I Cascade structures, the type I-C crRNA appears to have the closest helical conformation to that of the type I-D crRNA[31,36]. We resolved 43 bases of the crRNA scaffold in the Cascade-dsDNA complex, as well as 16 bases of the target strand (TS) and 13 bases (five of which we modeled as poly-T) of the non-target strand (NTS). Overall, the type I-D complex is more structurally similar to type III than type I effector complexes.

**PAM recognition by Cas10d.** To elucidate the mechanism of PAM recognition during dsDNA binding, we performed electrophoretic mobility shift assays with probes containing a complementary protospacer and various 3-bp PAM sequences. Type I-D Cascade bound the 5′-GTT-3′ PAM-protospacer probe, consistent with our previous publications[25]. However, type I-D

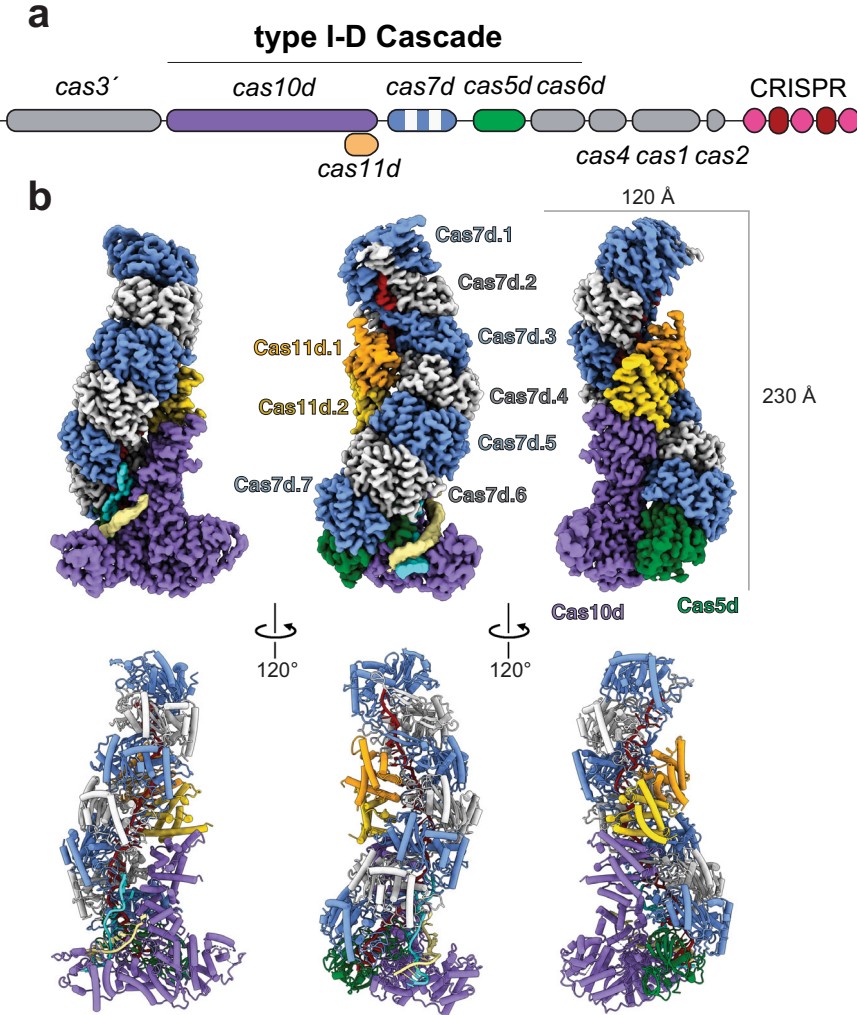

**Fig. 1 Architecture of type I-D Cascade. a** Schematic of the type I-D CRISPR-Cas operon. Subunits and nucleic acids are colored as follows: Cas10d, purple; Cas5d, green; Cas7, blue and white; Cas11d, gold and orange; TS, khaki; NTS, cyan; crRNA, red. Subunits not present in our structures are colored gray in the operon. **b** Cryo-electron microscopy reconstruction at 2.9 Å resolution (top) and corresponding atomic model (bottom) of type I-D Cascade. Components are colored as in **a**.

Cascade showed no detectable binding to a 5′-AAC-3′ PAM-protospacer and minimal binding to a 5′-CGT-3′ PAM-protospacer (Supplementary Fig. 1). This binding data shows that type I-D Cascade appears to bind dsDNA in a PAM-specific manner.

To understand the molecular basis for PAM recognition by type I-D Cascade, we analyzed specific interactions between Cas10d and the PAM duplex (5′-GTT-protospacer-3′ on the NTS and its complement on the TS) of the target dsDNA. The unwinding of the dsDNA duplex occurs between a cleft created by the bottom Cas7 subunit and the spur on Cas10d, which we assign as the PAM-recognition domain (PRD) (Fig. 2a). At the DNA duplex PAM site, we observe a loop from Cas10d wedged into the minor groove of the PAM duplex (Fig. 2b). One glycine (G433) interacts with the G (−3 position) in the NTS of the PAM, providing an anchor for the loop to lock into the minor groove (Fig. 2c). The wedge is further stabilized by hydrogen bonding between nearby residues Y437 and Q431 on Cas10d and the phosphodiester backbone of the PAM duplex (Fig. 2d). This is similar to the glycine-loop employed for PAM recognition in type I-E Cascade[37]. These interactions, along with many backbone-stabilizing interactions (Fig. 2c), aid in binding dsDNA targets. Interestingly, a well resolved Cas5 glutamine (Q110), which is

positioned where the dsDNA splits, and lysine (K114), which intercalates into the major groove of the PAM opposite to K326 from Cas10d, are close enough to interact with the A (−1 position) on the TS and the C (−3 position) of the NTS, respectively (Fig. 2c and Supplementary Fig. 6a). However, the Q110 residue likely only plays a role in non-specifically stabilizing the −1 position of the PAM, aiding PAM sliding. The K326 residue in Cas10d recognizes the C (−3 position) on the TS, the T (−2 position) on the NTS, and the G (−3 position) on the NTS via hydrogen bonding (Fig. 2e and Supplementary Fig. 6b). These results are consistent with previous bioinformatic and in vivo studies on type I-D PAM selection that showed these −3 and −2 PAM positions were the most important[29,30,35]. Together, these interactions suggest a process by which type I-D Cascade utilizes non-specific interactions for sliding along the DNA while the K326 finger scans for a GTN PAM.

To test the PAM recognition residues identified in our structure, we performed electrophoretic mobility shift assays with mutant type I-D Cascades and the dsDNA target probe described above. Size-exclusion chromatography indicated that the mutant Cascades profiles were nearly identical to wild-type complex, with a peak corresponding to ~420 kDa. SDS-PAGE analysis of the peak fractions showed the presence of all

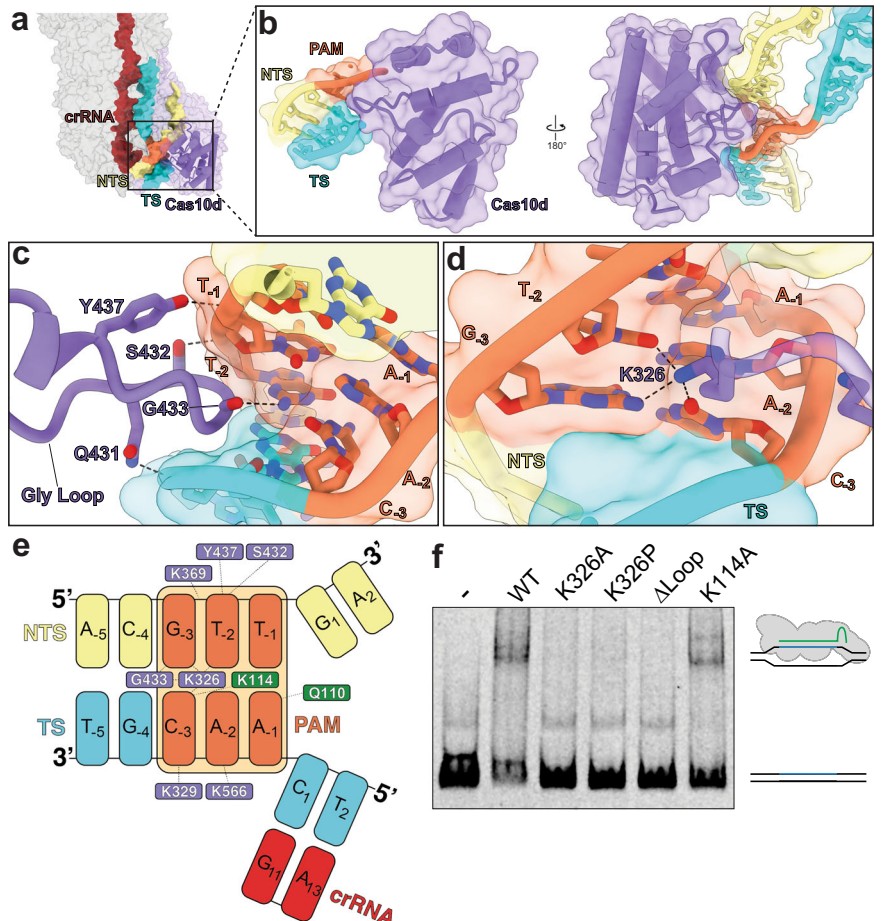

**Fig. 2 PAM recognition by Cas10d. a** R-loop formation in the type I-D complex. The TS travels up along the Cas7 filament, while the NTS travels across the face of Cas10d. **b** PAM recognition by Cas10d. A glycine loop from the PAM recognition domain of Cas10d integrates into the minor groove of the PAM in the DNA duplex. **c** Residues responsible for stabilizing and recognizing PAM nucleotides. Eight residues were found to be within 3.5 Å of the phospho-diester backbone of the dsDNA target. **d** Alternate angle of the glycine loop locked in the minor groove of the PAM. **e** Lysine 326 of Cas10d hydrogen bonds with C12 of the target strand PAM and T5 of the non-target strand PAM. These correspond to positions 1 and 2 within the PAM. **f** Electrophoretic mobility shift assay of fluorescently labeled protospacer dsDNA with a 5′-GTT-3′ PAM incubated with wild-type I-D Cascade, or mutated complexes containing Cas10d(K326A), Cas10d(K326P), Cas10d(ΔLoop), or Cas5(K114A). Representative of three independent experiments. Source data are provided as a Source data file.

components of the effector (Supplementary Fig. 7). These results suggest that the mutant Cascades are intact and fully assembled complexes. Strikingly, a single mutation of the key PAM sensing residue K326 in Cas10d to either an alanine or a proline completely abolished binding to the dsDNA target (Fig. 2f). Similarly, truncation of residues S432-Y437 from the glycine loop (Δ loop) also prevented Cascade from binding to the probe. Mutation of K114 (K114A) in Cas5d had little to no effect on binding, suggesting that it plays either an accessory or stabilizing role (Fig. 2f). These results strongly support our structural model for PAM recognition by type I-D Cascade.

**Cas10d guides the NTS of DNA towards its HD site.** PAM recognition triggers DNA duplex unwinding, enabling the TS to hybridize with the crRNA and displace the NTS to create an R-loop. Within the type I-D structure, we observe six bases of the NTS that traverse the face of Cas10d after bifurcation from the TS (Fig. 3a). Identities of the bases could not be established from the cryo-EM map, so we modeled this portion of the NTS as a poly-T. In our model, the otherwise unstructured NTS is stabilized by a highly conserved patch of positively charged residues (Fig. 3b). Notably, we see contiguous density between R680 and the NTS,

likely signaling NTS backbone stabilization (Supplementary Fig. 6c). However, this region of the NTS is not resolved well enough to highlight direct contacts. Contacts between this Cas10d patch and the NTS likely stabilize the nascent R-loop at early stages of TS:NTS duplex melting and thus favor R-loop completion via kinetic partitioning[9,38,39]. While this patch of positively charged residues is somewhat analogous to the K-vise and K-rim found in type I-E and I-F Cascade complexes[17,33], we observe a different path for the NTS in type I-D Cascade. We hypothesize that Cas10d directs the NTS towards its HD nuclease domain active site before rejoining the duplex at the top of the complex (Fig. 3c).

Multiple sequence alignment indicates high conservation scores for H81, H115, D116, and D210 of Cas10d, underscoring their putative role as a functional nuclease (Cas3″)[25]. Indeed, these four conserved residues are coordinated together in our structure in the HD domain of Cas10d and are consistent with other known HD sites in type I and III systems (Fig. 3d and Supplementary Fig. 8)[26,28]. The metal-coordinating histidines, H81 and H115, are on the N-terminal alpha-helices of the HD domain, which is a hallmark of the Cas3 HD domain (Supplementary Fig. 9). We observed continuous density between H81, H115, and D210, representing coordination of one divalent

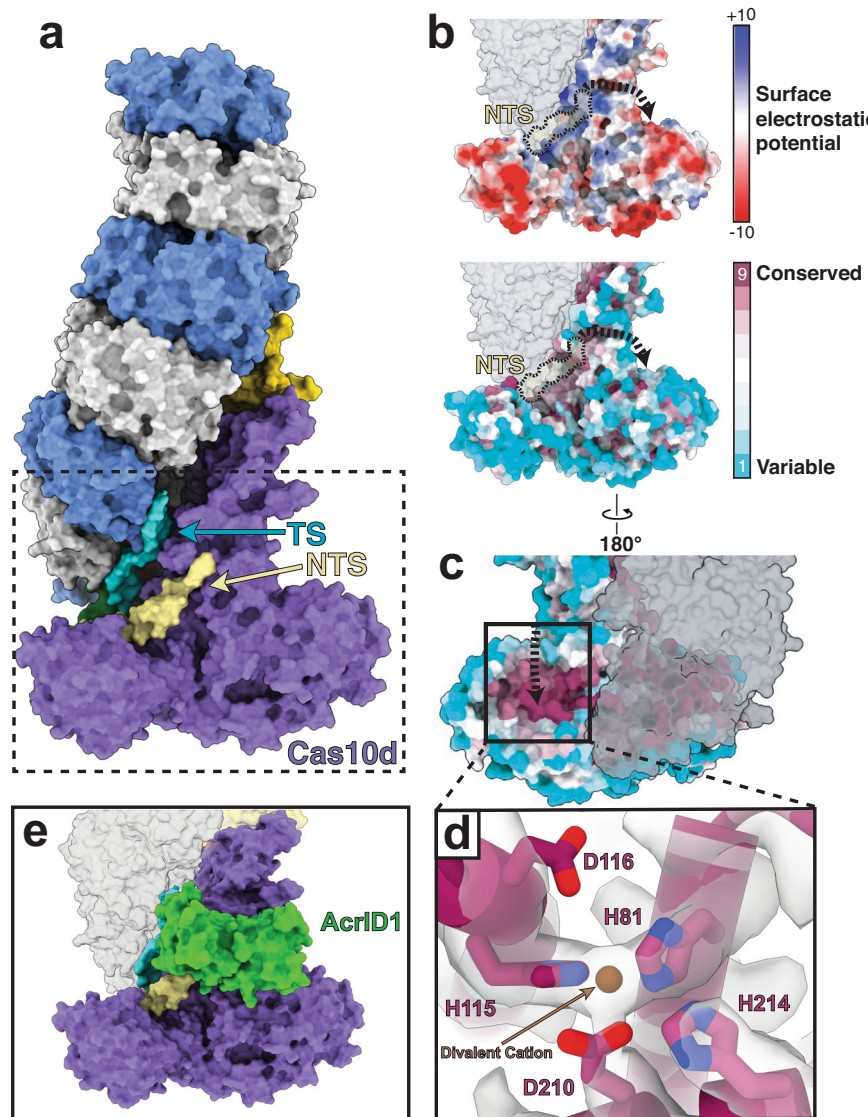

**Fig. 3 R-loop formation directs the non-target strand toward the HD domain. a** R-loop formation begins with unwinding of the PAM-proximal duplex with the target strand (cyan) traveling up along the Cas7 filament and the non-target strand traveling across the front of Cas10d. **b** Electrostatic surface potential and conservation scores appear correlated, where positive patches are more highly conserved and coordinate the non-target strand towards the active site on the back of Cas10d. Dashed line (black) represents the putative path of the NTS towards the active site of Cas10d, where nicking occurs. **c** Active site of Cas10d is highly conserved relative to the rest of the Cas10d surface. **d** H115, D116, H81, D210, and H214 form the active site of Cas10d to cleave the non-target strand. These four residues are highly conserved and likely coordinate a divalent cation. **e** Crystal structure of AcrID1 (PDB: 6thh, (Manav et al.[28])) overlayed onto the Cas10d subunit of our cryo-EM structure. AcrID1 prevents dsDNA R-loop formation by binding to the face of Cas10d, obstructing the path of the non-target strand towards the HD site. Subunits are colored as in Fig. 1.

cation. There was no observed density for residue D116, which indicates side chain flexibility. Interestingly, we also observe a third histidine (H214) pointing towards the active site, but it is unclear whether it plays a role in NTS degradation. The type I-D structure provides a clue about the mechanism whereby NTS DNA is poised proximal to the HD domain active site and primed to activate DNA degradation and create a Cas3′ helicase docking site[28]. This type I-D structure reveals how Cas10d likely achieves NTS DNA cleavage upon Cas3′ recruitment by providing a conserved, positively charged path to guide the NTS from the PRD toward the HD domain active site.

Recent studies have provided insights into how Cas10d from *Sulfolobus islandicus* interacts with the SIRV3 AcrID1 anti-CRISPR protein, which this virus uses to overcome targeting by type I-D Cascade. X-ray crystallography of the Cas10d-AcrID1 complex showed AcrID1 dimerizes before binding to one face of

Cas10d[28]. However, the mechanism for how this anti-CRISPR protein inhibited dsDNA degradation remained unclear. To identify the putative inhibitory mechanism of AcrID1, we overlaid the crystal structure of the Cas10d-AcrID1 complex onto the type I-D dsDNA-bound Cascade structure (Z-score 16.6, RMSD 15.32 across all pairs, 1.36 for pruned pairs) (Fig. 3e). Interestingly, the location of AcrID1 indicates that it competitively inhibits dsDNA binding by blocking the path of the NTS as it travels to the HD active site of Cas10d (Fig. 3e). Based on this structural analysis, there is no indication that AcrID1 binding to type I-D Cascade would affect single-stranded nucleic acid binding, consistent with SIRV3 being a dsDNA virus.

**PAM recognition triggers long-range conformation changes in Cas10d.** Previous evolutionary and genetic analyses have shown the similarities between type I-D Cascade and type III

complexes[7,25]. Later biochemical analyses of bacterial and archaeal type I-D Cascade complexes showed binding and cleavage of both dsDNA and ssDNA targets[25,34]. Although the biochemical analysis of type I-D Cascade nuclease activity on the TS and NTS provided insight into how archaeal type I-D Cascade degrades different targets, the mechanistic detail of how these targets are recognized remains unkown. It also remains enigmatic how type I-D Cascade selects the correct degradation pathway for each nucleic acid substrate.

Previous studies have shown that type I-D Cascade binds to ssDNA with high affinity[34]. However, because type I-D has an overall architecture and crRNA geometry that most closely resembles a type III-A complex, we speculated type I-D could also bind ssRNA. To investigate ssRNA binding and single-stranded vs double-stranded target discrimination by type I-D Cascade, we bound type I-D to a ssRNA target containing a protospacer that matched the crRNA spacer and performed cryo-electron microscopy. Our efforts resulted in a structure of the ssRNA-bound complex at 3.1 Å resolution (Fig. 4c). Overall, the structure is strikingly similar to the double-stranded target-bound complex (Figs. 1b and 4e). However, in the single-stranded target-bound complex, we lose recognizable density for the PAM recognition domain, likely due to the flexibility of this subunit in the presence of a single-stranded nucleic acid. We propose that the lack of a distinct dsDNA PAM in the ssRNA allows the PAM recognition domain to remain unlocked and inactive (PRD disengaged) (Figs. 4c, f and 5). Conversely, when PAM recognition occurs in double-stranded targets, the PRD locks into place (PRD engaged) (Figs. 4e, g and 5).

To better understand the mechanistic differences between single-stranded and double-stranded target selection, we segmented the Cas10d into three separate domains (Cas11d domain, Cas3″ HD domain, and PAM-recognition domain, Fig. 4a, b) and visualized conformational changes between the ssRNA-bound and dsDNA-bound Cascade using a modevector map (Fig. 4d). This map highlighted dramatic conformational rearrangements within the Cas11d domain and the Cas3″ domain. In comparison to the ssRNA target-bound complex, the dsDNA-bound complex undergoes a continuous 5 Å shift of the small subunits away from the belly of the complex and slightly downwards towards Cas10d, along with a similar shift of Cas10d downwards. We propose that upon PAM recognition, the PRD pulls the rest of Cas10d away from the Cas11d and Cas7d subunits. This conformational change supports a model where the Cas11d subunits are pulled into a position relative to the Cas7d filament that both supports R-loop stability[25] and opening of a path for the NTS towards the Cas10d active site. These two structures suggest that the conformational differences between the PRD-active and PRD-inactive states are vital for NTS cleavage. Our structural studies revealed a flexible PRD in the ssRNA-bound complex, which suggests that the type I-D Cascade does not require PRD engagement with a specific protospacer flanking sequence for ssRNA binding. Indeed, binding of a ssRNA substrate containing either a 5′-AAC-3′ PFS (corresponding to the complement of the GTT PAM) or a scrambled PFS (5′-ACG-3′) showed nearly identical binding with an apparent affinity of $11.0 \pm 0.9$ nM (Supplementary Fig. 10). We observed no binding to a non-specific control ssRNA that lacked complementarity to the crRNA.

We next investigated whether the structure could provide insights into single-stranded nuclease activity by type I-D Cascade. Type III effector complexes bind ssRNA and the Cas7 subunits perform periodic cleavage involving an aspartate residue in a loop region of the Cas7 palm domain, at a site adjacent to where the thumb domain intersects the crRNA:ssRNA duplex[36,40–42]. Interestingly, the Cas7d subunit from type I-D Cascade contains both sequence and structural homology to type III Cas7 subunits[6,43]. A recent study showed type I-D Cascade from *S. islandicus*[34] bound and cleaved ssDNA via its Cas7 subunits (SiCas7d)[34]; however, the mechanism was not clear. Lin and colleagues showed the loop of the palm domain of SiCas7d was not responsible for ssDNA cleavage but did show an E182Q mutation in the thumb domain reduced cleavage by 75%. We threaded the SiCas7d sequence onto our structure and found SiCas7d residue E182 aligned with *Synechocystis* Cas7d residue E152 (Supplementary Fig. 5b–d). Neither E182 of the SiCas7d threaded structure nor E152 of *Synechocystis* Cas7d are in a position compatible with target strand (TS) ssDNA cleavage due to the distance to the TS phosphodiester kink, indicating it is not directly part of the active site, though may still play a supportive role in ssDNA cleavage. Furthermore, from our structure, we observed the loop region in the palm domain was up to 16 Å from the TS, too far to support cleavage. Future studies are required to determine the exact mechanism of single-stranded nucleic acid cleavage.

## Discussion

Structures of multiple type I Cascade complexes have been determined, revealing the mechanisms underlying PAM recognition, R-loop formation, and Cas3 recruitment. Here, we solved the structures of type I-D Cascade bound to dsDNA and ssRNA, which revealed its chimeric nature and mechanistic insight into substrate selection. Type I-D Cascade exhibits a novel structure, given that its large subunit differs from the typical type I Cas8 subunit. Instead, the type I-D large subunit is a fusion of the type I Cas3 nuclease domain to an inactivated type III Cas10 subunit. Type I-D Cascade also has a similar crRNA-Cas7 helical arrangement to type III systems. These similarities highlight how type I-D Cascade represents an evolutionary hybrid between type I systems and a common type III-like ancestor.

Type I complexes scan for a PAM on dsDNA before using their crRNA to hybridize with a complementary sequence adjacent to the PAM, which results in Cas3 recruitment and processive degradation of the target DNA. PAM detection allows the complexes to differentiate self vs non-self DNA and leads to avoidance of CRISPR array targeting. Type III effector complexes instead depend on a lack of complementarity between the 5′ tag region of the crRNA and the ssRNA target to convert Cas10 from an autoinhibited state to a highly dynamic active state capable of non-specifically cleaving ssDNA proximal to the HD active site[26,44,45]. We observe that the activation mechanism of Cas10d carries hallmarks of both. Akin to type I systems, the process of GTN PAM recognition initiates crRNA:TS duplex formation, which triggers a long-range conformational change in Cas11d subunits, Cas3″, and Cas11d domain of Cas10d. While reminiscent of the activation mechanism of Cas10 from type III systems[44], the structural rearrangements within Cas10d triggered by PAM recognition are considerably larger than those observed for type III Cas10 subunits[26]. Upon PAM recognition, the Cas3″ domain becomes more ordered, suggesting that activation reduces Cas10d dynamics (Figs. 4 and 5). The active, locked Cas3″ domain is potentially a more suitable 'landing pad' for Cas3′ to dock, triggering rapid DNA unwinding and degradation. Together, this data indicates that while similar to type III systems, Cas10d provides a unique mode of activation for the type I-D system, consistent with an inactive Palm domain for cyclic oligoadenylate synthesis and downstream immune signaling in type III systems[28]. Interestingly, the type I-D system commonly co-occurs with type III systems[46], and indeed *Synechocystis* also encodes two type III CRISPR-Cas systems. It may be advantageous for the type I-D system to lack cOA synthesis activity to

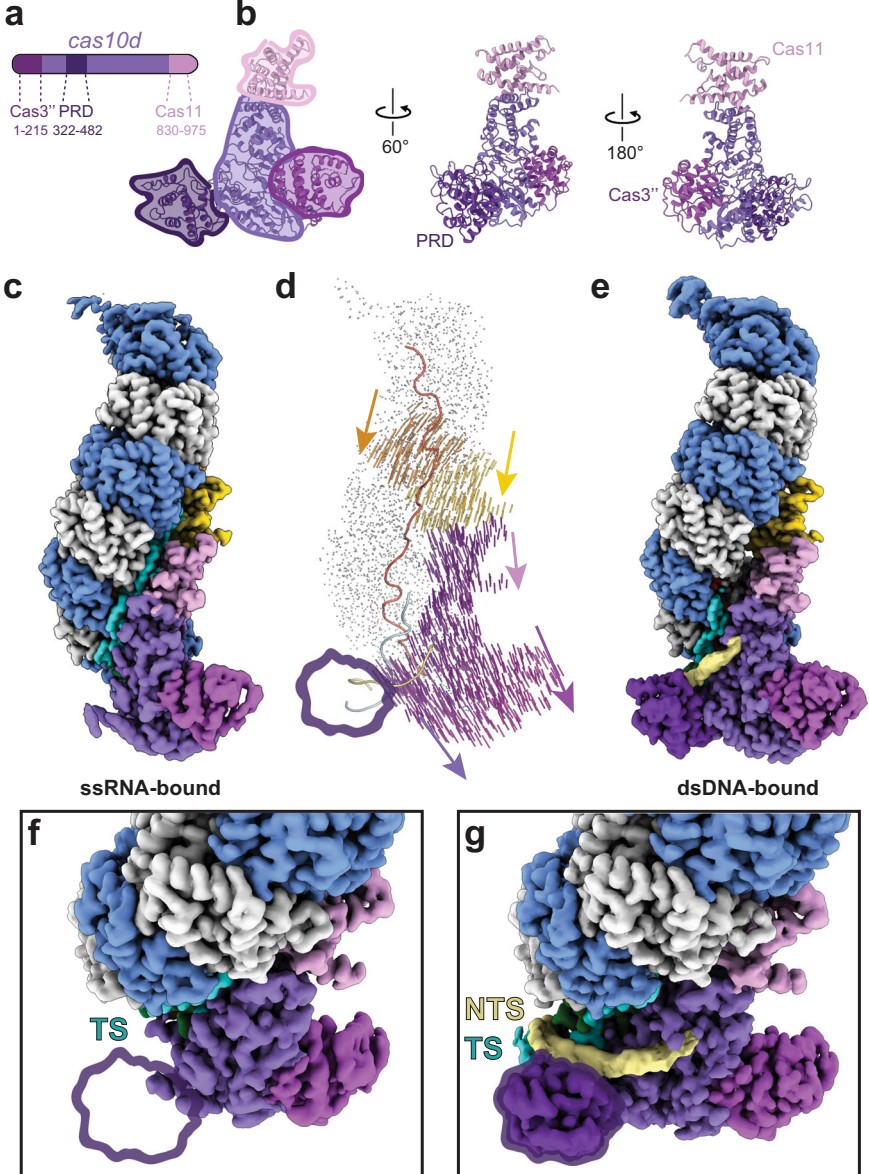

**Fig. 4 Conformational changes of I-D Cascade and PRD locking upon PAM recognition. a** Domain organization of Cas10d. Cas10d has a Cas3″ nuclease domain at the N-terminus (Cas3″, purple-maroon), a small subunit (Cas11) domain embedded within the C-terminus (Cas11, pink), and a PAM recognition domain (PRD, dark purple). **b** Structure of Cas10d. Domains are colored as in **a**. **c** Structure of type I-D complex bound to target ssRNA. There is no discernable density for the PRD (PRD-inactive). **d** Vector shift between conformations with and without PAM-recognition. Upon PAM recognition, the Cas11d and Cas10d subunits appear to shift downward ~5 Å. **e** Structure of type I-D complex upon PAM-dependent dsDNA target binding shows the PRD becomes ordered (PRD engaged). **f** Top-down view of PRD disengaged conformation. Without PAM recognition, the PRD density disappears at a higher threshold and the NTS is not present. **g** Top−down view of PRD engaged conformation. PRD density is visible and NTS density becomes disordered after the PAM. Subunits are colored as in Fig. 1 with Cas10d colored by domain.

avoid triggering downstream signaling pathways in trans upon activation.

While the type I-D structure resolved the path of the NTS across Cas10d away from the PAM-recognition domain, the definitive mechanism of Cas10d NTS nicking activity remains elusive. We did not observe nucleotide density within the HD active site of Cas10d. Likewise, other reported structures of type III complexes also lack detectable substrate in the Cas3″ HD domain[19,26,42,44]. Nevertheless, the trajectory of the NTS in the dsDNA-bound structure must deviate to the Cas3″ HD domain within Cas10d before returning to the top of the complex (Fig. 5), unlike other type I structures (O'Brien et. al, 2020; Chowdhury et. al, 2017; Hayes et. al, 2016). We hypothesize that conformational

changes to Cas10d upon PAM recognition create a platform for Cas3′ recruitment to type I-D Cascade, which in turn stabilizes the NTS to be positioned within the HD domain active site and permits cleavage. This model is consistent with results by Lin and colleagues that only observed cleavage of the NTS from dsDNA in the presence of Cas3′[34]. In other type I CRISPR systems, Cas3 association with Cascade occurs as a transient intermediate. The Cascade-Cas3 complex can disassociate during processive dsDNA degradation, depending on tension within the DNA[13]. However, even upon Cascade dissociation, Cas3 can still translocate and degrade the DNA target for several kilobases[13,47]. Since type I-D systems have the Cas3″ HD domain fused to Cas10d and the Cas3′ helicase is separately encoded, it is likely that in order to

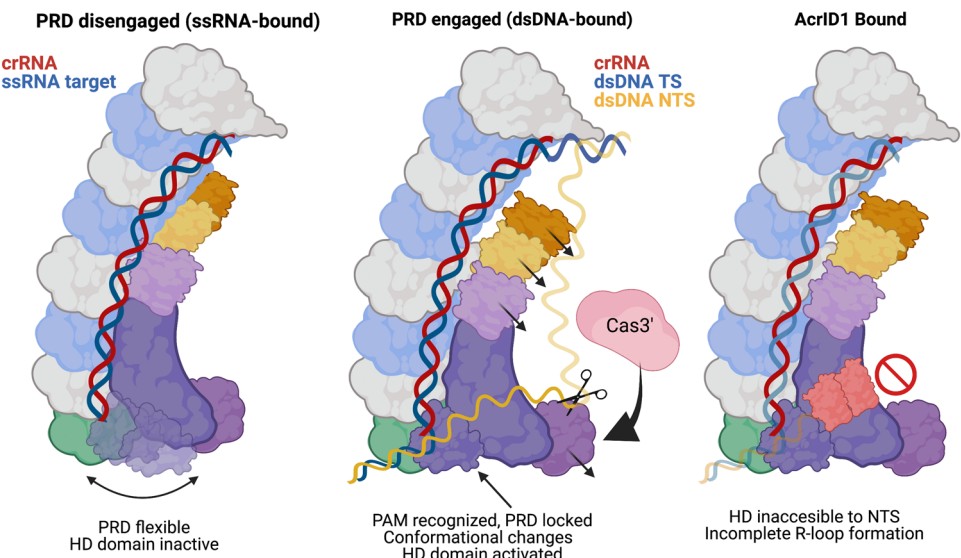

**Fig. 5 Model for type I-D Cascade activity.** Single-stranded target-bound Cascade has a flexible PRD (PRD disengaged). Upon binding of a dsDNA target, the PRD engages, recognizing the PAM and locking it into place. This allows I-D Cascade to undergo conformational rearrangements in Cas11d and Cas10d that allow for activation of the HD domain for cleavage of the NTS. The binding of AcrID1 blocks the path of the non-target strand towards the Cas10d HD site, preventing binding and R-loop formation with dsDNA targets. The nucleic acids are colored and labeled above each cartoon. Subunits are colored as in Fig. 1.

achieve processive degradation, the Cas3′:Cascade complex must remain intact.

We observed that the PAM-recognition domain (PRD) of Cas10d was highly flexible in the absence of a target duplex. This flexibility may be optimal for PAM scanning, allowing type I-D Cascade to rapidly probe DNA sequences. In contrast, upon PAM recognition and DNA duplex capture, the PRD adopts a more rigid, defined position to engage the rest of the degradation machinery. While type I-F Cascade contains a beta-hairpin that has been proposed to act as a "duplex splitter" by wedging between the TS and NTS upon PAM recognition, we do not observe such a feature in type I-D Cascade. Nevertheless, we propose a mechanism whereby PAM recognition leads to the PRD becoming locked in a static conformation, thus holding the TS:NTS duplex proximal to the exposed bases of the crRNA (Fig. 5). This would suggest that productive capture of PAM-proximal TS bases in Watson-Crick interactions with the crRNA may be a consequence of the reduced entropy upon PRD locking, favoring spontaneous, transient DNA duplex opening. Parallel mechanisms have been proposed for RNA chaperone proteins containing large disordered domains, which have been proposed to use entropy transfer to decrease the energy barrier to destabilize metastable RNA secondary structures enabling refolding[48,49]. To prevent the DNA duplex from re-annealing during the initial stages of R-loop formation, the positively charged patch that stretches across the front of Cas10d may stabilize the NTS, akin to a boot strap. The cumulative effect of rapid TS base-capture and NTS stabilization is kinetically partitioned R-loop formation, reducing the sizable energy barrier required to melt the DNA duplex by favoring the forward reaction (i.e., R-loop formation) over the backward reaction (re-annealing of the TS:NTS duplex)[50,51].

Alignment of the previously determined *S. islandicus* Cas10-d:AcrID1 crystal structure with the complete type I-D Cascade model revealed that AcrID1 binds to the positively charged belly of the complex, which is required for NTS stabilization. This binding may explain the observed competitive inhibition of AcrID1 on dsDNA binding[28], as Cascade:AcrID1 bound with less affinity to target dsDNA, presumably to perform PAM scanning.

However, by occluding the NTS-stabilizing surface of Cas10d, we propose that AcrID1 may prevent R-loop completion and block the path of the NTS to the HD domain active site, ultimately preventing Cas3′ recruitment and DNA degradation.

Type I-D Cascade can specifically bind dsDNA, ssDNA, and ssRNA. This highlights that type I-D Cascade might have the potential to provide defense against a broad range of invading nucleic acids. Our results shed light on the mechanism of PAM recognition and substrate selection by type I-D Cascade. We reveal conformational rearrangements that occur within Cas10d that lead to complex activation. An open question is how type I-D Cascade can coordinate its choice of ssRNA, ssDNA, or dsDNA targets during interference.

## Methods

Refer to Supplementary Tables 1 and 2 for lists of all plasmids and oligonucleotides, respectively, used in this study.

**Culture conditions**. Refer to Supplementary Table 3 for a list of all strains used in this study.

Unless otherwise noted, Escherichia coli strains were grown at 37 °C in Lysogeny Broth (LB), or on LB-agar (LBA) plates with 1.5% (w/v) agar. Media were supplemented with antibiotics when required as follows: ampicillin (Ap; 100 µg/mL), chloramphenicol (Cm; 25 µg/mL), and kanamycin (Km; 50 µg/mL).

**Cascade purification for structural studies**. Type I-D Cascade was purified via an N-terminal His₆-Cas6d and N-terminal His₆-Cas10d from plasmids pPF1549 and pPF1552, as previously described in McBride et al.[25]. The exact sample from this prior publication was used for the cryo-EM structural studies presented here. Briefly, the cell pellet was resuspended in lysis buffer (50 mM HEPES-NaOH, pH 7.5, 300 mM KCl, 5% Glycerol, 1 mM DTT, 0.02 mg/mL DNaseI, cOmplete EDTA free protease (Roche), and 0.1 mM of PMSF) supplemented with 10 mM imidazole and cells lysed. The lysate was applied to a HisTrapTM HP (GE Healthcare) column equilibrated in binding buffer (10 mM HEPES-NaOH, pH 7.5, 300 mM KCl, 5% Glycerol, 1 mM DTT), eluted using a gradient against binding buffer containing 500 mM imidazole. The His₆ tags on Cas6d and Cas10d were removed by cleavage with TEV protease during overnight dialysis in SEC buffer. The liberated His₆ tags and non-specific *E. coli* proteins were removed using a second HisTrap affinity column and the flow through was collected. Size exclusion chromatography (SEC) separated free Cas6d and Cas10d from the complex on a HiLoad 16/600 Superdex 200 column equilibrated in SEC Buffer (10 mM HEPES-NaOH, pH 7.5, 100 mM KCl, 5% Glycerol, 1 mM DTT).

**Generation of plasmids and purification of Cascade for biochemical assays**. A plasmid (pPF2451) for a His$_{10}$ tag with an enhancer sequence, TEV protease recognition sequence, and linker region was constructed by annealing and extending two oligonucleotides (primers PF3653 + PF3654) by PCR and cloning the product into pACYCDuet-1 via NcoI and BamHI restriction sites. A plasmid intermediate (pPF2452) for expression of His$_{10}$ tagged-Cas5d, Cas6d, and Cas11d was constructed by PCR-amplifying *cas5d* and *cas6d* (primers PF4980 + PF4981) and the region that encodes Cas11d within *cas10d* (primers PF4982 + PF4983) using *Synechocystis* genomic DNA as a template and Gibson assembly to clone the products into plasmid pPF2451 via BamHI and HindIII restriction sites. A plasmid (pPF2453) for the expression of His$_{10}$-tagged-Cas5d, Cas6d, and Cas11d with the first spacer (5′-GATTGTTGTGCCCCTGGCGGTCGCTTTCAATGCCT-3′) and flanking repeat sequences (5′-CTTTCCTTCTACTAATCCCGGCGATCGGGACT GAAAC-3′) from the type I-D associated CRISPR array was constructed by PCR-amplifying from *Synechocystis* genomic DNA (primers PF2937 + PF2938) and cloning the product into pPF2452 via NdeI and KpnI restriction sites.

A plasmid (pPF2455) for the expression of Cas10d and Cas7d was constructed by PCR-amplifying their genes (primers PF4991 + PF4992) using *Synechocystis* genomic DNA as a template and Gibson cloning the product into pPF1719 via SphI and KpnI restriction sites.

Plasmid pPF1719 was constructed by ligating the AraC-pBAD promoter fragment resulting from the digestion of pSEVA1810 with KpnI and PstI with a similarly digested plasmid pSEVA251.

Plasmids pPF3021, pPF3023, pPF3026, and pPF3025 are for expression of mutants Cas10d(K326P), Cas10d(K326A), Cas10d(ΔS432-Y437), and Cas5d(K114A), respectively. Plasmids pPF3021, pPF3023, and pPF3026 were constructed by site-directed mutagenesis through amplifying plasmid pPF2455 with primers PF6197 + PF6198, PF5633 + PF5634, and PF6207 + PF6208, respectively. Plasmid pPF3025, was constructed by amplifying plasmid pPF2453 with primers PF6205 + PF6206. Each were treated with DpnI to remove PCR template, and Gibson assembly to ligate the PCR product into the mutated plasmid.

Type I-D Cascade with N-terminal His$_{10}$-Cas5d was expressed in LOBSTR cells containing plasmids pPF2453 and pPF2455 for wild-type complex, pPF2453 and pPF3021 for Cas10d(K326P) mutant complex, pPF2453 and pPF3023 for Cas10d(K326A) mutant complex, pPF2453 and pPF3026 for Cas10d(ΔS432-Y437) mutant complex, or pPF3025 and pPF2455 for Cas5d(K114A) mutant complex. Cultures were induced with 1 mM IPTG and 0.2% arabinose at OD$_{600}$ = 0.6 and grown overnight at 18 °C. Cells were harvested at 10,000 × *g* for 15 min. The cell pellet was resuspended in 20 mL of lysis buffer supplemented with 10 mM imidazole and cells lysed by French press at 10,000 psi. The lysate was clarified by centrifugation at 15,000 × *g* for 15 min and the lysate was applied to a HisTrap affinity column equilibrated in binding buffer and eluted against a gradient against binding buffer containing 500 mM imidazole. The His$_{10}$ tag on Cas5d was removed by cleavage with TEV protease during overnight dialysis at 4 °C in binding buffer. The liberated His$_{10}$ tag and non-specific *E. coli* proteins were removed using a second HisTrap affinity column and the flow through was collected. The sample was concentrated with a centrifugal concentrator (Amicon; 100 kDa molecular weight cut off (MWCO)) and further purified from free Cas5d by size exclusion chromatography (SEC) on a HiLoad 16/600 Superdex 200 (GE Healthcare) column equilibrated in SEC Buffer. Purified complexes were typically concentrated to 1.5 mg/mL using a centrifugal concentrator (Amicon; 100 kDa MWCO), aliquoted, and stored at −80 °C.

**Electrophoretic mobility shift assays**. A plasmid (pPF1609) carrying the protospacer of the type I-D CRISPR array spacer 1, flanked by a 5′-GTT-3′ PAM was constructed by ligating annealed oligonucleotides PF3089 + PF3090 into pPF1590 via SpeI and XhoI restriction sites. A plasmid (pPF1610) carrying the protospacer of the type I-D CRISPR array spacer 1, flanked by a 5′-AAC-3 PAM was constructed by ligating annealed oligonucleotides PF3091 and PF3092 into pPF1590 via SpeI and XhoI restriction sites.

The 153-bp 5′ and 3′ IRD700 fluorescently labeled dsDNA probes containing the complementary protospacer and PAM sequences 5′-GTT-3′ and 5′-AAC-3′ were amplified by PCR using primers PF3158 and PF3160 from template plasmids pPF1609 and pPF1610, respectively. The complementary protospacer with the 5′-CGT-3 PAM was amplified from a gBlock (primer PF5590) with PF4095 and PF4096, and the non-specific probe was amplified from pPF1590 using primers PF3158 and PF3160. Double-stranded DNA binding assays were performed with or without 400 nM type I-D Cascade (purified via His$_{10}$-Cas5 method). Cascade was incubated with 2.5 nM fluorescently labeled probes at 30 °C for 60 min in a total volume of 10 µL (final conditions: 10 mM HEPES-NaOH, pH 7.5, 100 mM KCl, 5% v/v glycerol, 1 mM DTT, 0.01% v/v triton X-100, 1 µg BSA, and 0.1 µg poly(dI.dC)). Final reactions were separated on 4% polyacrylamide (19:1 acrylamide:bisacrylamide) native gel containing 0.5× TBE at 4 °C. The fluorescent probe was imaged using the Odyssey Fc imaging system (LICOR), and the results were analyzed with the Image Studio Lite software.

Single-stranded RNA probes were 60-nucleotides with 5′ IRD800 fluorescent labels. The probes contained the protospacer sequence complementary to the crRNA spacer with the protospacer flanking sequence 5′-AAC-3′ (primer PF3167), 5′-ACG-3′ (primer PF5591), and 5′-GUU-3′ (primer PF3322). A non-specific probe (primer PF3079) with no complementarity to the crRNA was also used.

Single-stranded RNA binding assays were performed with increasing concentrations (0, 3, 4, 6, 8, 10, 14, 18, 24, 33 nM) of type I-D Cascade. Cascade was incubated with 5 nM fluorescently labeled probes at 30 °C for 60 min in a total volume of 10 µL (final conditions: 10 mM HEPES-NaOH, pH 7.5, 100 mM KCl, 5% v/v glycerol, 1 mM DTT, 0.01% v/v triton X-100, 1 µg BSA, and 0.26 µg *E. coli* tRNA). Final reactions were separated on 4% polyacrylamide (19:1 acrylamide:bisacrylamide) native gel containing 0.5× TBE at 4 °C. The fluorescent probe was imaged and analyzed as with the dsDNA, except on the 800 nm filter. The signals from bound and unbound ssRNA were quantified using Image Studio. Data were plotted on GraphPad Prism (version 8.0.1) and curve fitting was carried out by non-linear regression using one site-specific binding with Hill slope. The apparent binding dissociation constants for type I-D Cascade were determined from three independent experiments.

**Cryo-EM grid preparation and data collection**. We mixed type I-D Cascade with either ssRNA (primer PF4100) or dsDNA (DNA fragment duplex PF4099) at a 1:2 Cascade:target molar ratio. Target binding was facilitated by incubating the mixture at 30 °C for 30 min. For both datasets, 1.5 µL of the target-bound I-D Cascade was applied to each side of C-flat 4/2 holey carbon grids (Protochips Inc.) after plasma cleaning for 30 s with a Solarus 950 plasma cleaner (Gatan). The sample was vitrified using an FEI Vitrobot MarkIV kept at 4 °C and 100% humidity. For the ssRNA-bound dataset, two samples of 0.16 mg/mL and 0.08 mg/mL complex were applied to two separate grids and blotted for 4 s with a blot force of 0. For the dsDNA-bound dataset, a sample of 0.125 mg/mL dsDNA-bound complex was applied to a grid and blotted for 4.5 s with a blot force of 0. Data was acquired from each grid on an FEI Titan Krios (Sauer Structural Biology Lab, University of Texas at Austin) operating at 300 kV with a nominal magnification of 22,500× (1.045 Å/pixel). Complexes were exposed with a dose rate of 15e⁻/pixel/s for 3 s, leading to a total exposure of 45e⁻/pixel, and movies were collected over 20 frames (150 ms/frame) on a Gatan K3 direct electron detector. Images were collected using a defocus range of −1 to −2 µm for the ssRNA-bound sample and −1.2 to −2.2 µM for the dsDNA-bound sample. Data collection was automated using LEGINON[52]. A full description of the cryo-EM data collection parameters can be found in Supplementary Table 4.

**Cryo-EM data processing**. Motion correction, CTF estimation, and micrograph masking were performed on raw movies from the Krios using Warp v1.0.9[53]. Particles were picked and extracted using the neural network-based particle picker BoxNet in Warp and uploaded to cryoSPARC v3.2[54]. Particles from the two ssRNA-Cascade datasets were combined to yield a total of ~1.5 million particles. Additionally, ~2.6 million particles were selected from the dsDNA-Cascade micrographs.

After preprocessing the dsDNA-Cascade dataset, 2 rounds of 2D classification and filtering were done, filtering out ~1.4 million particles leaving a total of ~1.2 million particles for further analysis. Four rounds of 3D classification and subsequent heterogeneous refinement were done to filter out ~1 million more particles, leaving a total of ~257k final particles. After a non-uniform refinement[55] of this particle set using an initial model low-pass filtered to 30 Å from a heterogeneous refinement, we obtained a model with a nominal resolution of 3 Å according to the 0.143 FSC gold-standard. However, this model did not contain continuous, buildable density for Cas11d, the HD domain of Cas10d, and the PRD of Cas10d.

After preprocessing the ssRNA-Cascade dataset, we employed 4 rounds of 2D classification on ~1.5 million particles to filter out ~950k leaving a total of ~550k particles for further analysis. After 3D classification and subsequent heterogeneous refinement for 2 iterations, a non-uniform refinement was performed on a final dataset of ~163k particles, yielding a model with a nominal resolution of ~3.1 Å. Interestingly, we observed a nucleic acid duplex protruding out of the complex by the PRD for this ssRNA-Cascade dataset, similar to the dsDNA-Cascade model. We speculate that there was a co-purifying double-stranded nucleic acid within our complexes and reasoned that bound ssRNA complexes could be uncovered by focused classification on the Cas10d subunit. This would also address the issue of poor density in the Cas11d domain, the PAM-recognition domain, and the Cas3″ HD domain within Cas10d.

Initial processing of separate datasets resulted in reconstructions with many poorly resolved regions, precluding de novo model building. This is likely due to the high degree of conformational heterogeneity exhibited by the complex, and the abundance of Cas7 filaments within the sample. These Cas7 filaments are ubiquitous in Cascade samples and have been identified in previously published data of Cascades[15]. To overcome this, we chose to combine our datasets, and computationally "purify" our particles through focused classification on a region of interest that was otherwise very poorly resolved (namely Cas10d, since this showed the greatest degree of heterogeneity). This strategy enabled us to separate two distinct structures of I-D Cascade bound to single- and double-stranded nucleic acids. This approach was crucial for determining high-resolution, high-quality reconstructions and enabled us to model Cas10d de novo.

We began our focused classification tests by pooling the ssRNA-Cascade and dsDNA-Cascade datasets together, yielding a dataset of ~4.1 m total particles. We performed a Refine3D job in RELION v3.0 to align the particles for focused classification[56]. As expected, the model was of too low resolution to properly

estimate. We then performed focused 3D classification without alignments with 3 classes, again yielding indistinguishable models of resolution too low for proper estimation. These three classes contained ~850k, ~990k, and ~2.07 m particles, respectively, and were imported into cryoSPARC v3.2 for 2D classification and filtering, leaving ~650k, ~790k, and ~1.06 m particles for each class, respectively. We then re-pooled the particles and performed an ab initio reconstruction and 2 iterations of heterogeneous refinement and selected one class of ~900k particles. These particles were then aligned via non-uniform refinement, then re-added to RELION v3.0 for focused classification without alignments. We ran this focused classification using a regularization parameter (tau factor) of 20 and an initial model from the initial dsDNA-Cascade dataset low-pass filtered to 15 Å. We chose to classify into four classes, two of which generated models of ~4.1 Å resolution or better. Class 1 was reconstructed to ~3.7 Å resolution from a set of ~336k particles and appeared to have a strong density for both the PAM-recognition domain and the non-target strand. Class 2 contained a set of ~167k particles and appeared to have a very poor density for the PAM-recognition domain that disappeared at a high threshold. We further refined both using non-uniform refinement in cryoSPARC v3.2 yielding models of ~2.9 Å for the dsDNA-Cascade structure and ~3.1 Å for the ssRNA-Cascade structure. To aid in model building, we performed local B-factor sharpening using DeepEMhancer v20210511[57]. A simplified cryo-EM data processing workflow for the pooled particles is summarized in Supplementary Fig. 2.

**De novo model building and refinement**. Due to a lack of models with confidence values above 24% on the Phyre 2 server[58], model building for the dsDNA-Cascade structure was done entirely de novo. We used PSIPRED[59] to perform secondary structure predictions on each subunit within the complex before building secondary structure elements manually in Coot[60]. Coordinates were refined using ISOLDE v1.2 in ChimeraX v1.0[61–63] and comprehensive validation was performed in Phenix v1.18rc5[64] to analyze model quality. To build the ssRNA-Cascade structure, we used the ChimeraX fit-in-map function with the dsDNA-bound model, removed domains that lacked density, then performed Molecular Dynamics Flexible Fitting using Namdinator v20191016-5814c947[65]. Iterative rounds of model building and refinement of this model against the ssRNA-Cascade map were performed using Coot v0.8.9 and ISOLDE v1.2. Many of the data processing and refinement programs described above were organized by SBGrid[66]. Model statistics are provided in Supplementary Table 4.

**Multiple sequence alignments, conservation scores, and structural threading**. To analyze the conservation of residues in Cas10d, we performed a protein blast of the *Synechocystis* Cas5d peptide sequence and downloaded an alignment with gaps of the top 100 matches. We then measured the conservation of Cas10d residues using ConSurf[67].

When analyzing our Cas7d structure and potential target strand-cleaving residues, we used iTasser[68] to thread the *S. islandicus* Cas7d sequence onto our *Synechocystis* Cas7d structure. To further test whether we were analyzing the correct aspartate residues, we performed a sequence alignment between *S. islandicus* and *Synechocystis* Cas7d using T-Coffee[69–72].

**Reporting summary**. Further information on research design is available in the Nature Research Reporting Summary linked to this article.

## Data availability

The data that support this study are available from the corresponding author upon reasonable request. The cryo-EM structure and associated atomic model of dsDNA-bound Cascade have been deposited into the Electron Microscopy Data Bank and the Protein Data Bank with accession codes EMD-24974 and PDB 7SBA, respectively. The cryo-EM structure of ssRNA-bound Cascade and associated atomic model have been deposited with accession codes EMD-24976 and PDB 7SBB, respectively. Source data are provided with this paper.

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

## Acknowledgements

We thank R. O'Brien, J. Wright, and I. Strohkendl for comments and helpful discussions on the manuscript and A. Dai for expert cryo-EM assistance. We thank S. Cameron, University of Otago, for making the type I-D mutant constructs. Data were collected at the Sauer Structural Biology Laboratory at UT Austin. The model cartoon was created using Biorender. This work was supported in part by a Marsden Fund Fast-Start Grant (to R.D.F.) from the Royal Society of New Zealand (RSNZ), the Marsden Fund (to P.C.F.), Bioprotection Aotearoa (Tertiary Education Commission, NZ), and National Institute of General Medical Sciences (NIGMS) of the National Institutes of Health (NIH) R35GM138348 (to D.W.T.). T.M.M. was supported by the University of Otago Doctoral Scholarship. D.W.T is a CPRIT Scholar supported by the Cancer Prevention and Research Institute of Texas (RR160088).

## Author contributions

E.A.S. performed cryo-EM data collection, processing, and structure determination. E.A.S., J.P.K.B., and D.W. performed model building and analysis. T.M.M. and R.D.F. purified the complexes and T.M.M. performed biochemical assays. E.A.S., J.P.K.B., and D.W.T. wrote the manuscript with input from all authors. P.C.F., R.D.F., and D.W.T. conceived and supervised the studies and obtained funding for this work.

## Competing interests

R.D.F., P.C.F., and D.W.T. are inventors on a patent application based on this research titled "Type I-D CRISPR-Cas Systems and Uses Thereof," which has been assigned Australian Patent Application No. 2022900672, and the filing date of 18 March 2022. The remaining authors declare no competing interests.
