## [Peer Review File · Nature Communications]

Structural rearrangements allow nucleic acid discrimination by type I-D CascadeREVIEWER COMMENTS

Reviewer #1 (Remarks to the Author):

In this work Schwartz et al describe a structure-function analysis of the type I-D CRISPR-Cas system. Type I-D systems are particularly interesting as they contains features of both type I and type III systems, and are possibly and evolutionary link between these two types. However, we know comparatively little about the structure and mechanisms of type I-D systems. To address this gap in knowledge Schwartz et al determined the cryo-EM structures of type I-D Cascade bound to either a double-stranded (ds)DNA or a single-stranded (ss)RNA target. From these structures the authors note several potentially important interactions between Cascade subunits and the PAM-DNA and note a conformational change that likely primes Cascade for Cas3' binding. Finally, they present a modeling analysis that suggests how AcrID1 blocks target binding via competitive inhibition. The manuscript is well written and the figures are clear. However, I have some concerns about how the cryo-EM processing was done and the lack of experiments to establish the importance of observations from the structure. These concerns are outlined in the detail below (Major points). I also include a few other suggestions (Minor points). Addressing these should improve the impact and rigor of this study and the authors have all the required expertise and reagents to perform these additional experiments.

Major points:

I have concerns about how the cryo-EM processing was handled. Three datasets were collected. Two of the dsDNA-Cascade one of the ssRNA-Cascade. The authors state in their methods "Because we observed a co-purifying double-stranded nucleic acid-bound Cascade in the ssRNA dataset, we began focused classification tests by pooling the ssRNA-Cascade and dsDNA-Cascade datasets together...". I both fail to see the logic in this approach and don't understand what the advantage it could have, nor is any of this further explained. It's not clear what the double-stranded nucleic acid is in the ssRNA-Cascade dataset but it is unlikely that it will be recognized in the same way was as a specific target, for example, side chain interactions with the PAM. So what advantage does pooling the particles offer? At its worst this approach could limit the clarity in regions of the maps due to contaminating particles in the final reconstructions. In a revised manuscript the authors have two ways to handle this issue. Firstly, explain and justify why the particles were pooled in the manuscript text, not just is the response to the reviewers. I'm perfectly happy to be educated as to why this approach is justified. Secondly, repeat the cryo-EM processing without pooling the particles from different complexes. Note, that pooling the two datasets of the dsDNA-Cascade is of course reasonable.

The analysis of the structure is well performed and the resulting conclusions are reasonably compelling but are limited without accompanying mutagenesis/biochemical experiments. For example, several residues are identified as interacting with PAM but the importance of these residues to actual PAM

binding remains unknown. The authors have presented DNA binding data (Figure S1B and S8B) and so these follow-up experiments should be routine for the authors and would significantly increase the rigor and impact of these studies. Therefore a revised manuscript should include a mutagenesis/biochemical analysis of at least the more important interactions observed in the structure to establish their importance.

The only pictures of EM density shown are relatively big picture views of the whole complex (Figures 1B, 4C, 4E, S2, and S3F) or more close up views of the large sections of the complex (Figures 4F, 4G and S2). There are no pictures presented of the important interactions discussed in the text, including the PAM-DNA and the residues that interact with it. This makes it very difficult for the reader to evaluate the strength of conclusions drawn about these interactions. A revised manuscript must include pictures of the relevant density. It would also be helpful to show density pictures for each individual protein subunit, the crRNA and the DNA, maybe as a supplemental figure.

Minor Points:

Given the flexibility and/or heterogeneity at the top of the complex did the authors perform focused classifications to more clearly resolve this end of the complex (as they did for the other end) and perhaps resolve the Cas6d subunit?

The structures presented contain seven Cas7 subunits. The footprint of Cas7 is typically six nucleotides and therefore the Cas7 filament here should cover ~42-nt of RNA. However, the spacer of the crRNA used is 35-nt long. Could the authors comment how the “extra” Cas7 is accommodated on their crRNA? Does it perhaps bind to the repeat region of the crRNA?

Was Figure 5 generated with BioRender? If so then the appropriate reference should be added.

The initial particle images (no.) in supplementary table 4 is stated as 4,11,214 for both structures. This is misleading as this is the number of pooled particles.

Reviewer #2 (Remarks to the Author):

This manuscript reports the first high resolution structure of a type I-D Cascade structure, in complex with either ssRNA or dsDNA. Type I-D cascades are particularly interesting as they represent an

evolutionary link between type I and type III systems. The structure reveals important new details about the mechanism of dsDNA binding and PAM recognition. This work will be of broad interest and represents a significant step forwards in the field.

A significant weakness of the paper is the absence of key biochemical / mutagenesis data to back up the predictions made from the structure. In particular, the role of PAM interacting residues is not explored.

Major points:

1. The structure of a type I-D cascade bound to ssRNA is presented. A prediction would be that this type of target is cleaved by Cas7, as observed for ssDNA targets. After all, if ssRNA could bind but was not cleaved it could interfere with dsDNA targetting. Was this observed? This data would be easily generated if not available. The importance of this question is highlighted by the final sentence of the manuscript discussion.
2. Cas6 is not observed in the complex although it is suggested to co-purify. Cas6 in type I systems requires the presence of the 3' end of the crRNA, which it binds to – often in a hairpin conformation. Is the 3' end of the crRNA also missing in these structures, and is this due to flexibility or degradation?
3. Residues postulated to play a key role in PAM recognition are described. However, there is no biochemical data to test these predictions. Can this be obtained and added? The authors do present biochemical data for PAM dependent DNA binding in the supplemental data, so they have the assays working and it should not be hard to generate this data.
4. Figure 5 shows a model for type I-D cascade activity. The “No PAM” state on the left is a bit confusing for the reader. What is shown here is the structure bound to ssRNA. It is very unlikely that “no PAM” dsDNA will ever form a complex.
5. A second problem with figure 5 is that it appears to show only 6 cas7 subunits rather than seven. Is one hidden at the back of the structure? Some more description of the subunit colour scheme here would help too.

Specific points:

1. Line 63. Many type III systems lack the HD nuclease domain altogether. For those that do, the evidence of nicking of transcription bubbles is very weak and not supported by all the references used here (18-21).
2. Line 149. Two residues are mentioned (Q110 and K114) but here only one is highlighted – please clarify.
3. Line 155 – should “sliding across” read “sliding along”? The former suggests moving perpendicular to the DNA helix.

4. Line 197. The suggestion here that ssDNA cleavage by Cas10d is relevant only for ssDNA viruses is misleading – all dsDNA viruses do generate ssDNA, most commonly during replication, which could generate targets for this activity.

5. Line 312. It may be worth mentioning here that some type III systems have a “split” Cas3 as well as possessing an HD nuclease domain in Cas10.

Reviewer #3 (Remarks to the Author):

Structural rearrangements allow nucleic acid discrimination by type I-D Cascade

Schwartz et al.

Type I and III are among the most prevalent CRISPR-Cas systems. A major difference between them is their target nucleic acid: dsDNA for type I, and ssRNA for type III. Type I-D stands out by containing elements that are characteristic of both types. The functional consequences of this hybrid architecture were not clear, partly due to the absence of structural information on this group of CRISPR-Cas systems.

In this manuscript, Schwartz and colleagues present cryo-EM structures of *Synechocystis* Type I-D Cascade bound to either dsDNA or ssRNA. The structures are of high quality, with resolutions around 3Å and reveal a significant conformational change between the ssRNA- and dsDNA-bound structures. This change is accompanied by an ordering of the PAM-recognition domain (PRD). Another important insight of the structural analysis presented is the proposal of an electrostatic path that could guide the single-stranded non-template strand (NTS) from its branching point at the PAM towards the putative nuclease site in Cas10d's HD domain for cleavage.

This is a well-written, concise manuscript that presents important structures in a very active area of research. Cascade systems are of much interest both biologically and biotechnologically. Thus, a structural and mechanistic understanding of the unique properties of any system will be of interest to a broad audience.

My only concerns about the work relate to some of the hypotheses proposed. It seems to me that they could be tested relatively easily, yet they were left as speculation. Adding at least some of those experiments to a revised version of the manuscript would strengthen it significantly. I list my major and minor concerns below.

Major concerns:

- One of the major mechanistic hypotheses proposed in the manuscript is that a conserved, basic surface guides the NTS towards the active site in Cas10d's HD domain. It seems to me that this is a relatively simple idea to test by making a couple of mutations that change the electrostatics of that surface and testing cleavage rates. I would not take a negative result as an indication that the authors' hypothesis is incorrect (there are many reasons why the experiment may not work), but this is not a reason not to try. A positive result would significantly bolster their idea.
- Related to the point above: I assume the authors looked and did not find anything, but I was wondering if any traces of the NTS could be found going towards the putative active site of Cas10d at lower thresholds, or even by doing some focused classification in this area.
- The other hypothesis regards the active site of Cas10d's HD domain. There are a couple of issues here. First, the authors mention in the text that they see continuous density among three of the residues there (H81, H115 and D210), and that this is an indication of a divalent cation being coordinated. A statement like this requires a figure showing the part of the map being discussed, potentially even showing stronger density for the putative cation. This is too big a structural point to make without a supporting figure. Second, the putative catalytic nature of this center should also be easy to test with just one mutant and an assay. This would also strengthen the manuscript considerably.

Minor concerns:

- The authors state that "... we were unable to resolve Cas6d... although it was observed in the purified complex via SDS-PAGE and mass spectrometry". However, only a citation is given for this. To make this claim here, they need to show SDS-PAGE/MS data for the sample used to obtain the cryo-EM structures presented. If the sample is the same one that was tested in reference (25), this should be stated explicitly.
- The colored outline for the NTS in Fig.3b eclipses the most important part of this panel: the areas of high positive electrostatic potential and high conservation. I would suggest keeping the dashed outline (maybe make it thicker so it remains visible?) but remove the semi-transparent coloring so readers can see the colors below.
- Fig.5a: Should the blue strand not be absent from the panel in the left? This is supposed to be the single-stranded target-bound Cascade. (It makes it additionally confusing when a single-stranded nucleic acid appears in the middle panel, now annealed to a suddenly longer blue strand.)
- Supplementary Fig. 5a. This panel is referred to in the text when discussing direct contacts that R680 makes with the NTS. However, the figure does not show any contacts. It would help to display the NTS in atomic representation, indicating those contacts (as is the case in panel (b)).
- Supplementary Fig. 8b: The legend refers to the second PFS as 5'-ACG but the figure itself shows 5'-UCG. The other issue is the coloring referring to the different PFS's. The gradient bars above the gels match what the legend says, but the colors of the substrates themselves (on the left of the gels) are the same for all three. It would probably help readers if the colors of the substrates matched the gradients and the legend.

- At the end of page 9 there is a reference to “(Fig. 4,6)”. Since there is no Fig.6, I am not sure what this was meant to be.
- In page 10, the claim “(W)hile the type I-D structure resolved the path of the NTS across Cas10d towards the HD domain active site” seems a bit of a stretch to me. I agree that the structure suggests that, and a relatively simple experiment could further support the idea as suggested above, but the structure could only “resolve” the path if density for the NTS had actually been seen reaching the HD domain’s active site. This should be toned down a bit.

Response to Editor and Reviewers

We thank the reviewers for their rigorous and constructive feedback. We have addressed their comments in the point-by-point response below. In general, based on their suggestions, we have improved the clarity of the text and figures and/or figure legends and explained our attempts at identifying Cas6 in the complex using focused classification. We have also performed biochemical assays to interrogate effects of mutating important regions identified in PAM binding within our structure. We believe that these changes based on comments from the reviewers have significantly strengthened the manuscript.

Reviewer #1

Major Points:

1. I have concerns about how the cryo-EM processing was handled. Three datasets were collected. Two of the dsDNA-Cascade one of the ssRNA-Cascade. The authors state in their methods “Because we observed a co-purifying double-stranded nucleic acid-bound Cascade in the ssRNA dataset, we began focused classification tests by pooling the ssRNA-Cascade and dsDNA-Cascade datasets together...”. I both fail to see the logic in this approach and don’t understand what the advantage it could have, nor is any of this further explained. It’s not clear what the double-stranded nucleic acid is in the ssRNA-Cascade dataset but it is unlikely that it will be recognized in the same way as a specific target, for example, side chain interactions with the PAM. So what advantage does pooling the particles offer? At its worst this approach could limit the clarity in regions of the maps due to contaminating particles in the final reconstructions. In a revised manuscript the authors have two ways to handle this issue. Firstly, explain and justify why the particles were pooled in the manuscript text, not just is the response to the reviewers. I’m perfectly happy to be educated as to why this approach is justified. Secondly, repeat the cryo-EM processing without pooling the particles from different complexes. Note, that pooling the two datasets of the dsDNA-Cascade is of course reasonable.

Initial processing of separate datasets resulted in reconstructions with many poorly resolved regions, precluding de novo model building. This is likely due to the high degree of conformational heterogeneity exhibited by the complex, and the abundance of Cas7 filaments within the sample. These Cas7 filaments are ubiquitous in Cascade samples and have been identified in previously published data of Cascades (Hochstrasser and Taylor, 2016). To overcome this, we chose to combine our datasets, and computationally “purify” our particles through focused classification on a region of interest that was otherwise very poorly resolved (namely Cas10d, since this showed the greatest degree of heterogeneity).

This strategy enabled us to separate two distinct structures of I-D Cascade bound to single- and double-stranded nucleic acids. This approach was crucial for determining high-resolution, high-quality reconstructions and enabled us to model Cas10d de novo.

This rationale has now been added to the manuscript on at the bottom of pg. 16.

Combining datasets and separating distinct conformations is a standard and well-established approach. Here are two great examples (of many):

<https://www.nature.com/articles/s41586-020-2087-1>

Pooled particles from multiple datasets containing different ATP analogues were used to improve the resolution of the overall complex to allow for de-novo modelling.

<https://www.sciencedirect.com/science/article/pii/S1097276521001313>

Pooled particles from multiple previously published datasets to enable visualization of an important, low abundance, structural intermediate at high resolution.

This has been added to the manuscript on p. 18

2. The analysis of the structure is well performed and the resulting conclusions are reasonably compelling but are limited without accompanying mutagenesis/biochemical experiments. For example, several residues are identified as interacting with PAM but the importance of these residues to actual PAM binding remains unknown. The authors have presented DNA binding data (Figure S1B and S8B) and so these follow-up experiments should be routine for the authors and would significantly increase the rigor and impact of these studies. Therefore a revised manuscript should include a mutagenesis/biochemical analysis of at least the more important interactions observed in the structure to establish their importance.

We have now performed binding assays with Cas10d mutants. We chose to focus on the regions highlighted in Fig. 2c and 2d. We observed that both removal of the glycine loop (Δ loop) and mutation of K326 prevented binding of dsDNA by I-D Cascade, supporting our hypotheses.

This data is now included in Fig. 2f, and the following text has been added to the results section of the manuscript on pg 6:

“To test the PAM recognition residues identified in our structure, we performed electrophoretic mobility shift assays with mutant type I-D Cascades and the dsDNA target probe described above. Size-exclusion chromatography indicated that the mutant Cascades profiles were nearly identical to wild type complex, with a peak corresponding to ~420 kDa. SDS-PAGE analysis of the peak fractions showed the presence of all components of the effector (Supplementary Fig. 7). These results suggest that the mutant Cascades are intact and fully assembled complexes. Strikingly, a single mutation of the key PAM sensing residue K326 in Cas10d to either an alanine or a proline completely abolished binding to the dsDNA target. Similarly, truncation of residues S432-Y437 from the glycine loop (Δ loop) also prevented Cascade from binding to the probe. Mutation of K114 (K114A) in Cas5d had little to no effect on binding, suggesting that it plays either an accessory or stabilizing role. These results strongly support our structural model for PAM recognition by type I-D Cascade.”

We have also included the methods for mutant plasmid construction and complex expression, in addition to the required primers and plasmids.

3. The only pictures of EM density shown are relatively big picture views of the whole complex (Figures 1B, 4C, 4E, S2, and S3F) or more close up views of the large sections of the complex (Figures 4F, 4G and S2). There are no pictures presented of the important interactions discussed in the text, including the PAM-DNA and the residues that interact with it. This makes it very difficult for the reader to evaluate the strength of conclusions drawn about these interactions. A revised manuscript must include pictures of the relevant density. It would also be helpful to show density pictures for each individual protein subunit, the crRNA and the DNA, maybe as a supplemental figure.

To show the quality of the reconstruction, we've created a new Supplementary Fig. 4, which includes the full map with the docked model, density of each subunit, and a representative beta strand in Cas7d.

We have also updated Fig. 3 to show the putative divalent cation, as well as the nearly contiguous density between it and several of our proposed active site residues. We've also added a panel to Supplementary Fig. 6 to show the EM density for K326.

Minor Points:

1. Given the flexibility and/or heterogeneity at the top of the complex did the authors perform focused classifications to more clearly resolve this end of the complex (as they did for the other end) and perhaps resolve the Cas6d subunit?

We tried both processing the data as described in the methods with focused classification at the top of the complex, as well as processing the ssRNA-bound and dsDNA-bound datasets separately with focused classification. We identified a class with a hard cutoff at 8 Cas7 subunits with very poor density for the 8th Cas7 and a class with what appears to be continuous density that becomes less interpretable as it continues up the complex, likely due to Cas7 filaments.

2. The structures presented contain seven Cas7 subunits. The footprint of Cas7 is typically six nucleotides and therefore the Cas7 filament here should cover ~42-nt of RNA. However, the spacer of the crRNA used is 35-nt long. Could the authors comment how the “extra” Cas7 is accommodated on their crRNA? Does it perhaps bind to the repeat region of the crRNA?

The 72-nt crRNA consists of a 35-nt spacer flanked by 8-nt 5' handle and 29-nt 3' repeat. The 3' repeat region is expected to have a 15-nt ssRNA stretch before the stem-loop (Scholz *et al.* 2013). In our structure, we observe seven Cas7 subunits bound to 46-nts of the crRNA. We anticipate an 8th Cas7 subunit could bind before steric hinderance from Cas6 at the stem-loop would limit further Cas7 assembly. We have subsequently performed a focused classification of this region of the complex and revealed a class with eight Cas7 subunits, but with very poor density for the 8th Cas7, signaling heterogeneity.

3. Was Figure 5 generated with BioRender? If so then the appropriate reference should be added.

According to BioRender's website, a citation that reads “Created with BioRender.com” can be placed in the acknowledgements, amongst other places. This is where we chose to place the reference. Here is the text from BioRender.com

'Whether you're publishing in a journal, textbook, or simply in a presentation or departmental website, all users must cite BioRender figures with the credit “Created with BioRender.com.” You can include this wherever it makes sense, like the figure caption, citations list, or acknowledgments.'

4. The initial particle images (no.) in supplementary table 4 is stated as 4,11,214 for both structures. This is misleading as this is the number of pooled particles.

We used this data to obtain the two structures shown in the manuscript. We believe it is best practice to be as clear as possible about the fact that both structures came from the same initial 4,111,214 particle set.

Reviewer #2:

Major Points:

1. The structure of a type I-D cascade bound to ssRNA is presented. A prediction would be that this type of target is cleaved by Cas7, as observed for ssDNA targets. After all, if ssRNA could bind but was not cleaved it could interfere with dsDNA targeting. Was this observed? This data would be easily generated if not available. The importance of this question is highlighted by the final sentence of the manuscript discussion.

Based on the binding assay in Supplementary Fig. 10, we do not observe ssRNA cleavage by I-D Cascade. To our knowledge, specific ssRNA cleavage by Cas7d has not been reported by other groups. This is the case in Hrlé et al., 2014, where ssRNA binding by Cas7d was measured, and in Lin et al., 2020 where ssRNA binding by the entire I-D Cascade was investigated. We did not observe TS cleavage in our high-resolution ssRNA-bound structure. We believe that this is beyond the scope of our manuscript and would require significant further investigation.

The notion that ssRNA binding without cleavage might interfere with dsDNA targeting is interesting, given the high affinity for RNA and higher copy number of RNA transcripts compared to DNA genome. However, we don't believe interference studies in either I-D or I-E (which also demonstrated ssRNA binding) have addressed this specific question. The underlying mechanism of substrate choice by Cascade is an interesting and open question that we hope to address in future studies, hence our choice of final sentence in the Discussion.

2. Cas6 is not observed in the complex although it is suggested to co-purify. Cas6 in type I systems requires the presence of the 3' end of the crRNA, which it binds to – often in a hairpin conformation. Is the 3' end of the crRNA also missing in these structures, and is this due to flexibility or degradation?

Our data suggests that we are not able to resolve the predicted full mature crRNA. We attempted particle re-centering and focused classification at the top of the complex but could not resolve Cas6d in the complex. We did, however, reconstruct a class with 8 Cas7d subunits present. We believe the lack of Cas6 density is likely due to both heterogeneity and flexibility at the top of the complex. While a 3' hairpin is predicted, we do not observe a hairpin in any of the classes.

3. Residues postulated to play a key role in PAM recognition are described. However, there is no biochemical data to test these predictions. Can this be obtained and added? The authors do present biochemical data for PAM dependent DNA binding in the supplemental data, so they have the assays working and it should not be hard to generate this data.

This is an excellent point. In the revised manuscript, we report biochemical experiments that address this concern. It is presented in Fig. 2f. Below is the text that has been added to the revised manuscript to address this point on pg. 6:

“To test the PAM recognition residues identified in our structure, we performed electrophoretic mobility shift assays with mutant type I-D Cascades and the dsDNA target probe described above. Size-exclusion chromatography indicated that the mutant Cascades profiles were nearly identical to wild type complex, with a peak corresponding to ~420 kDa. SDS-PAGE analysis of the peak fractions showed the presence of all components of the effector (Supplementary Fig. 7). These results suggest that the mutant Cascades are intact and fully assembled complexes. Strikingly, a single mutation of the key PAM sensing residue K326 in Cas10d to either an alanine or a proline

completely abolished binding to the dsDNA target. Similarly, truncation of residues S432-Y437 from the glycine loop (Δ loop) also prevented Cascade from binding to the probe. Mutation of K114 (K114A) in Cas5d had little to no effect on binding, suggesting that it plays either an accessory or stabilizing role. These results strongly support our structural model for PAM recognition by type I-D Cascade.”

4. Figure 5 shows a model for type I-D cascade activity. The “No PAM” state on the left is a bit confusing for the reader. What is shown here is the structure bound to ssRNA. It is very unlikely that “no PAM” dsDNA will ever form a complex.

Thanks for pointing out this oversight. We have changed the language to “PRD engaged (dsDNA-bound)” and “PRD disengaged (ssRNA-bound)” to improve clarity.

5. A second problem with figure 5 is that it appears to show only 6 cas7 subunits rather than seven. Is one hidden at the back of the structure? Some more description of the subunit colour scheme here would help too.

Again, thank you for pointing out this oversight. Fig. 5 has been updated to show 7 Cas7 subunits.

Minor Comments:

1. Line 63. Many type III systems lack the HD nuclease domain altogether. For those that do, the evidence of nicking of transcription bubbles is very weak and not supported by all the references used here (18-21).

We agree with the reviewer. We have adjusted the wording of this sentence to reflect that this is merely a hypothesis:

“It was hypothesized the type III signature protein, Cas10, may induce non-specific ssDNA nicks at the transcription bubble¹⁸⁻²¹.”

2. Line 149. Two residues are mentioned (Q110 and K114) but here only one is highlighted – please clarify.

This has been fixed and is now included in panel e of Fig. 2.

3. Line 155 – should “sliding across” read “sliding along”? The former suggests moving perpendicular to the DNA helix.

Great point. It now reads as “sliding along.”

4. Line 197. The suggestion here that ssDNA cleavage by Cas10d is relevant only for ssDNA viruses is misleading – all dsDNA viruses do generate ssDNA, most commonly during replication, which could generate targets for this activity.

We do not make any claims that ssDNA cleavage by Cas10d is relevant only for ssDNA viruses. We do not have any evidence of Cas10d activation based on PFS matching, so our only hypothesis of Cas10d activation is based on dsDNA binding and subsequent cleavage of the NTS by Cas10d. This is all activated by PAM recognition. We also claim that we don't anticipate AcrID1

affects single stranded nucleic acid targets hybridizing with the crRNA, as Cas10d is not involved in this hybridization.

5. Line 312. It may be worth mentioning here that some type III systems have a “split” Cas3 as well as possessing an HD nuclease domain in Cas10.

This is surprising, since Cas3 is the signature subunit for type I systems. While the type III-like systems that contain a Cas3 exist, this is not very understood, and we would rather not comment on it at this time.

Reviewer #3:

Major Points:

1. One of the major mechanistic hypotheses proposed in the manuscript is that a conserved, basic surface guides the NTS towards the active site in Cas10d's HD domain. It seems to me that this is a relatively simple idea to test by making a couple of mutations that change the electrostatics of that surface and testing cleavage rates. I would not take a negative result as an indication that the authors' hypothesis is incorrect (there are many reasons why the experiment may not work), but this is not a reason not to try. A positive result would significantly bolster their idea.

We expressed and purified the full complex containing an R680E mutation in Cas10d. This mutant appeared to bind the same dsDNA target with equal affinity to the WT complex. We propose this is likely due to the cumulative effect of multiple positive residues that stabilize the NTS. Expressing and purifying the mutants that we present was already challenging due to pandemic regulations in each of our respective laboratories, so it would be quite difficult to mutate a large patch.

2. Related to the point above: I assume the authors looked and did not find anything, but I was wondering if any traces of the NTS could be found going towards the putative active site of Cas10d at lower thresholds, or even by doing some focused classification in this area.

Unfortunately, the mask we used for focused classification included the entirety of Cas10d, and we could not resolve more of the NTS. This is likely due to continuous flexibility of the NTS around the Cas10d active site, which is not amenable to structure determination.

3. The other hypothesis regards the active site of Cas10d's HD domain. There are a couple of issues here. First, the authors mention in the text that they see continuous density among three of the residues there (H81, H115 and D210), and that this is an indication of a divalent cation being coordinated. A statement like this requires a figure showing the part of the map being discussed, potentially even showing stronger density for the putative cation. This is too big a structural point to make without a supporting figure. Second, the putative catalytic nature of this center should also be easy to test with just one mutant and an assay. This would also strengthen the manuscript considerably.

The HD domain activity of type I-D Cascade is already well-established (as shown in Lin et al., 2020). This study also demonstrated that mutations of the HD domain render type I-D Cascade unable to cleave dsDNA. This group also showed that the Cas3' helicase subunit was required for cleavage of dsDNA by the HD domain using *in vitro* cleavage assays.

We have found that purification of *Synechocystis* Cas3' is highly challenging. We have dedicated significant time, effort and resources to this (including fusions with various solubility tags), but we have not been able to express and purify this subunit. Therefore, we believe that pursuing experiments involving Cas3' activity (as required for dsDNA cleavage) is beyond the scope of our manuscript.

However, we have updated this figure to show the putative divalent cation, as well as the contiguous density. To reflect the importance of this point, this is now included within the main text as Fig. 3D.

Minor Comments:

1. The authors state that "... we were unable to resolve Cas6d... although it was observed in the purified complex via SDS-PAGE and mass spectrometry". However, only a citation is given for this. To make this claim here, they need to show SDS-PAGE/MS data for the sample used to obtain the cryo-EM structures presented. If the sample is the same one that was tested in reference (25), this should be stated explicitly.

The Cascade used for structural studies in this paper was the exact same sample for McBride *et al.*, 2020. We agree with the reviewer that we could be more explicit in acknowledging this. To address this, we have made changes to the results and the methods:

On pg. 4: "...we were unable to resolve Cas6d in this high-resolution structure, despite being the same sample where Cas6d was observed in the purified complex via SDS-PAGE and mass spectrometry²⁵."

On pg 13: "Cascade purification for structural studies

Type I-D Cascade was purified via an N-terminal His₆-Cas6d and N-terminal His₆-Cas10d from plasmids pPF1549 and pPF1552, as previously described in McBride *et al.*²⁵ The exact sample from this prior publication was used for the cryo-EM structural studies presented here. Briefly, the cell pellet was resuspended..."

2. The colored outline for the NTS in Fig.3b eclipses the most important part of this panel: the areas of high positive electrostatic potential and high conservation. I would suggest keeping the dashed outline (maybe make it thicker so it remains visible?) but remove the semi-transparent coloring so readers can see the colors below.

We have increased the transparency of the NTS in panel b of Fig. 3. We hope that this improves the clarity of presentation.

3. Fig.5a: Should the blue strand not be absent from the panel in the left? This is supposed to be the single-stranded target-bound Cascade. (It makes it additionally confusing when a single-stranded nucleic acid appears in the middle panel, now annealed to a suddenly longer blue strand.)

We have added labels to Fig. 5 to improve clarity. While I-D Cascade in the absence of target nucleic acids probably has a flexible PRD, we have decided to depict what we have observed in our data, where the PRD is flexible when bound to ssRNA but becomes locked when bound to dsDNA. We believe the revised legend and diagram are now clear.

4. Supplementary Fig. 5a. This panel is referred to in the text when discussing direct contacts that R680 makes with the NTS. However, the figure does not show any contacts. It would help to display the NTS in atomic representation, indicating those contacts (as is the case in panel (b)).

The reason we did not draw contacts here is because we don't have high enough confidence in the distances between NTS backbone and the R680 side chain. The NTS is very low resolution in this model (~5 Å), and thus we did not find it appropriate for us to draw contacts. We have updated the manuscript to convey this point:

“Notably, we see contiguous density between R680 and the NTS, likely signaling NTS backbone stabilization (Supplementary Fig. 6c). However, this region of the NTS is not resolved well enough to highlight direct contacts.”

5. Supplementary Fig. 8b: The legend refers to the second PFS as 5'-ACG but the figure itself shows 5'-UCG. The other issue is the coloring referring to the different PFS's. The gradient bars above the gels match what the legend says, but the colors of the substrates themselves (on the left of the gels) are the same for all three. It would probably help readers if the colors of the substrates matched the gradients and the legend.

Thank you for bringing this oversight to our attention. This figure has been updated to have the correct PFS sequence (5'-ACG). We have added more details to the figure legend to differentiate binding curves.

6. At the end of page 9 there is a reference to “(Fig. 4,6)”. Since there is no Fig.6, I am not sure what this was meant to be.

This has been fixed in the text. It now references Fig. 4,5.

7. In page 10, the claim “(W)hile the type I-D structure resolved the path of the NTS across Cas10d towards the HD domain active site” seems a bit of a stretch to me. I agree that the structure suggests that, and a relatively simple experiment could further support the idea as suggested above, but the structure could only “resolve” the path if density for the NTS had actually been seen reaching the HD domain's active site. This should be toned down a bit.

We recognize the confusing nature of this claim. We have updated this sentence in the manuscript to read “away from the PAM-recognition domain” rather than “towards the HD domain active site.”

REVIEWERS' COMMENTS

Reviewer #1 (Remarks to the Author):

The authors have largely addressed all the concerns raised in the previous round of reviews. The manuscript is much improved and ready for publication.

Reviewer #2 (Remarks to the Author):

The authors have dealt constructively with my comments and provided key new experimental data to support their conclusions.

Reviewer #3 (Remarks to the Author):

I am generally satisfied with how the authors have addressed my concerns in this revised manuscript and I support publication in Nature Communications, with one minor point to be addressed (below).

The authors mention in their rebuttal that they tested the R680E mutation in Cas10d and saw no effect in binding to the target dsDNA. I agree with their interpretation that this may be due to the cumulative effect of positively charged residues, and for that reason I don't think it is necessary to show this new data. I do think, however, that the authors need to tone down the statement in Supplementary Fig. 6c, where they claim that "R680 of Cas10d contacts the NTS backbone to guide the NTS towards the Cas10d HD site". A very minor change (e.g. saying something like "R680 is one of the residues involved in guiding...") that implies that other residues are also involved would suffice. Otherwise, the statement may give readers the idea that R680 is THE residue that guides the NTS, which cannot be the case given the result with R680E.

Response to Reviews

I do think, however, that the authors need to tone down the statement in Supplementary Fig. 6c, where they claim that "R680 of Cas10d contacts the NTS backbone to guide the NTS towards the Cas10d HD site". A very minor change (e.g. saying something like "R680 is one of the residues involved in guiding...") that implies that other residues are also involved would suffice. Otherwise, the statement may give readers the idea that R680 is THE residue that guides the NTS, which cannot be the case given the result with R680E.

We thank the reviewer for addressing this important point. We have edited the sentence to read:

R680 of Cas10d is one of the residues that contacts the NTS backbone to guide the NTS towards the Cas10d HD site.